# Stem Cell Transplantation and Cell-Free Treatment for Periodontal Regeneration

**DOI:** 10.3390/ijms23031011

**Published:** 2022-01-18

**Authors:** Kengo Iwasaki, Yihao Peng, Ryuhei Kanda, Makoto Umeda, Isao Ishikawa

**Affiliations:** 1Institute of Dental Research, Osaka Dental University, Osaka 573-1121, Japan; kanda-r@cc.osaka-dent.ac.jp; 2Department of Periodontology, Osaka Dental University, Osaka 573-1121, Japan; pengyh910806@gmail.com (Y.P.); umeda-m@cc.osaka-dent.ac.jp (M.U.); 3Institute of Advanced Biomedical Engineering and Science, Tokyo Women’s Medical University, Tokyo 162-8666, Japan; ishikawa.isao@twmu.ac.jp

**Keywords:** periodontal disease, regeneration, cell transplantation, conditioned medium, exosome

## Abstract

Increasing attention has been paid to cell-based medicines. Many in vivo and in vitro studies have demonstrated the efficacy of stem cell transplantation for the regeneration of periodontal tissues over the past 20 years. Although positive evidence has accumulated regarding periodontal regeneration using stem cells, the exact mechanism of tissue regeneration is still largely unknown. This review outlines the practicality and emerging problems of stem cell transplantation therapy for periodontal regeneration. In addition, possible solutions to these problems and cell-free treatment are discussed.

## 1. Introduction

Periodontal disease is a chronic inflammatory state of periodontal tissues caused mainly by the colonization of Gram-negative bacteria [1,2]. In the local area affected by periodontitis, chronic inflammation over a long period of time leads to progressive destruction of the tooth-supporting tissues, and eventually, the affected teeth are extracted due to the loss of supporting tissues [3].

Periodontal disease is one of the primary causes of tooth extraction worldwide [4], and it is widely recognized that periodontal treatment is important in maintaining oral functions including mastication and pronunciation. In addition, recent studies have reported that periodontal disease is related to the pathogenesis of various systemic diseases such as diabetes, cardiovascular disease, low birth weight, osteoporosis, etc. [5,6]. The importance of periodontal treatment has been emphasized from the perspective of maintaining systemic health.

The ultimate treatment goal of periodontal disease is to regenerate lost periodontal tissues including cementum, periodontal ligament, and alveolar bone. However, there is still no complete periodontal regenerative treatment, and thus, research is being conducted to develop a new treatment for periodontal regeneration. In this context, attempts to regenerate periodontal tissues by stem cell transplantation have been made over the past 20 years. A number of studies have examined whether various stem cells can regenerate periodontal tissues in multiple animal models. This review provides an overview of periodontal regeneration using stem cells and summarizes the characteristics, problems, and new research developments in periodontal regeneration using stem cells.

## 2. Periodontal Regeneration by Stem Cell Transplantation

Recent advances in cell biology and tissue engineering technologies have led to the development of therapeutic strategies to treat diseases by transplanting or administering cells cultured outside the human body [7,8]. In the field of regenerative medicine, the strategy of reshaping tissues that have lost their function by transplanting cells with the capacity to form these tissues is very reasonable. For the regeneration of periodontal tissues, various types of stem cells have been examined for their regenerative potential at the animal experimental level (Table 1). Of the many stem cell types, mesenchymal stem cells (MSCs) are the most frequently tested for periodontal regeneration. MSCs were originally identified from bone-marrow aspirates as a plastic-adherent fibroblastic cell population with multi-differentiation potential mainly into mesenchymal linages [9].

MSCs derived from a variety of tissues, including the bone marrow [10,11,12,13,14,15,16,17,18,19,20,21,22], fat [43,44,45], periodontal ligament [16,23,24,25,26,27,28,29,30,31,32,33,34,35,36,37,38,39,40], dental pulp [30,41,42], and gingiva [47,48], were transplanted into experimental animals. There are also reports of periosteal cells [16,46] and stem cells from human exfoliated deciduous teeth (SHED) [36]. Dogs, minipigs, sheep, mice, and rats were used as experimental animals, and the periodontal defect morphology was highly variable. The results of these preclinical animal studies have been validated by systematic reviews [49,50,51,52,53,54] and meta-analyses [55,56,57,58]. All meta-analyses concluded that stem cell transplantation is likely to induce the regeneration of periodontal tissues, although further studies at a larger scale with appropriate experimental design are needed.

Human clinical studies have already begun based on these animal studies. The results of the controlled clinical trial in human patients are summarized in Table 2 [59,60,61,62,63]. All of these studies confirmed the safety of stem cell transplantation and reported that clinical parameters such as pocket depth, clinical attachment level, and radiographic bone volume tended to improve by stem cell transplantation [59,60,61,62,63]. However, of the five meta-analyses, two found statistically significant differences between control and cell treatment. The results from human clinical studies seems to be less impactful than those from preclinical studies. Further controlled clinical studies with larger sample size and long term observation period are needed.

## 3. Problems in Periodontal Regenerative Therapy Using Stem Cell Transplantation

As mentioned above, many animal studies have demonstrated through histologic evaluation that periodontal tissue regeneration can be achieved by stem cell transplantation, suggesting that stem cell transplantation may be a promising new treatment. However, there are still some problems that remain unresolved.

### 3.1. Mechanism of Tissue Regeneration

In most stem cell-based tissue regeneration procedures, transplanted stem cells are expected to localize or migrate, proliferate, differentiate, and form new tissues at the transplanted site. However, many cell transplantation studies have reported that the actual engraftment of transplanted cells is very low [64,65,66,67]. In periodontal regeneration, very few studies have observed the long-term distribution of transplanted stem cells in regenerated periodontal tissues. Hasegawa et al. transplanted green fluorescent protein (GFP)-labeled bone marrow-derived MSCs into dogs with periodontal defects and showed that osteoblasts, cementoblasts, osteocytes, and fibroblasts in the regenerated tissues were GFP-positive after 4 weeks [11]. Tsumanuma et al. reported that RNA and DNA for enhanced GFP (EGFP) were detected in tissue specimens collected from dogs at 8 weeks after transplantation of allogenic periodontal ligament-derived MSCs with EGFP [39]. These results indicate the presence of transplanted cells in regenerated periodontal tissues; however, the extent to which they contribute to tissue regeneration remains unclear. Yu et al. transplanted GFP-labelled periodontal ligament stem cells (PDLSCs) into periodontal defects, examined the distribution of GFP-positive cells in tissue sections, and reported that GFP-positive PDLSCs exist in clumps in some areas and rarely became bone and osteoblasts [68]. Iwasaki et al. transplanted PKH26-labeled human PDLSCs into immunodeficient rats and examined the PKH26 signal in tissue sections 4 weeks later. PKH26-positive cells were observed as clumps in a small portion of the regenerated tissues [40]. In addition, they attempted to detect human-specific DNA using genomic DNA recovered from sections prepared 3, 7, and 28 days after surgery in the same experiment. The results showed that human DNA detection did not increase significantly in the tissues but rather decreased compared with immediately after the cell transplantation. These results suggest that transplanted stem cells are partially engrafted in regenerated tissues; however, their direct contribution to new tissue formation is limited. The local conditions after transplantation surgery are difficult for the survival of transplanted cells due to ischemia caused by vascular disruption and inflammatory reactions related to wound healing, which may limit the viability of the transplanted cells. In addition, in almost all animal experiments, regenerated tissues were formed in continuity with the surrounding tissues. This indicates that the regeneration of periodontal tissues by stem cell transplantation may be the result of accelerated spontaneous wound healing. If the transplanted stem cells are unlikely to engraft, proliferate, and differentiate to form tissue directly, the mechanism by which transplanted stem cells lead to tissue regeneration remains unclear.

### 3.2. Limitations of Autologous Stem Cells

In animal experiments, allogenic and xenogeneic cells are often used to examine their regenerative potential. However, when considering cell therapy in humans, the first choice is to transplant autologous cells. Autologous stem cell transplantation in patients with periodontal disease requires several considerations. The first is the limitation of tissue retrieval for harvesting stem cells. In order to culture stem cells, tissues such as the periodontal ligament, dental pulp, gingiva, bone marrow, and fat need to be harvested; however, these tissues are sometimes difficult to collect due to their scarcity. Second, the majority of patients with periodontal disease are middle-aged or older, and the effects of aging on tissues and cells need to be taken into account when transplanting autologous cells. Aging is known to negatively affect the number and quality of tissue stem cells [69,70]. Animal experiments are usually performed using stem cells from relatively young adult animals or young human volunteers. Animal experiments mimicking periodontal patients in age have not yet been conducted. Furthermore, ex vivo expansion of stem cells is necessary to ensure the number of cells required for cell transplantation. Stem cells from middle-aged and older donors are susceptible to replicative aging in culture, which may significantly impair their stem cell properties [71]. When considering autologous stem cell transplantation in patients with periodontal disease, available cell sources may be limited.

### 3.3. Safety Issues in Stem Cell Transplantation

The safety of stem cell transplantation for periodontal regeneration has been reported by many clinical studies [59,60,61,62,63,72]. Safety considerations regarding autologous stem cell transplantation include the transmission of infection during ex vivo expansion, and tumor or unintended tissue formation after transplantation. The number of cells required for cell transplantation may vary depending on the degree of periodontal tissue destruction; however, it is believed that approximately 10^6^ to 10^7^ cells are required per deep periodontal defect. Therefore, ex vivo expansion of cells is unavoidable in the stem cell treatment of periodontal disease. The contamination of pathogenic factors in the process of cell culture must be avoided. Therefore, GMP-compliant facilities are required to ensure a high level of safety, and at the same time, a variety of quality control inspections are essential. The required safety level for cell treatment in dentistry is the same as that of general cell therapy in the medical field. Cell production methods for human clinical use have been largely established, and as long as they are followed, there may be no concerns about safety.

In general, MSCs are thought to have very low tumorigenicity; however, it is assumed that, through cell culture and passaging, the cells are greatly affected by cellular stress, which increases the risk of tumorigenesis. Since most animal experiments were completed in a relatively short period of time, from several weeks to months, there is a paucity of data showing the long-term safety of MSC transplantation. Dannan et al. reported the development of squamous epithelial-cell carcinoma after transplantation of human periodontal ligament-derived stem cells into periodontal defects in athymic rats [73]. They found that, after transplantation, tumors were found in two out of four rats at 6 weeks and in all four mice at 8 weeks, and human mitochondria and karyotypic variation were observed in the tumors. These results may be very rare cases of animal experiments; however, the safety of MSC transplantation is not completely guaranteed. Stem cells can differentiate into various types of cells, and after transplantation into the human body, we cannot control their spontaneous growth and differentiation. Therefore, there is a possibility of unintended tissue formation by transplanted stem cells. Safety has not been established in this regard because of the lack of detailed long-term histological studies of the fate of transplanted cells.

### 3.4. Cost of Cell Therapy

It is not clear what the reasonable treatment cost is for cell-based dental treatments. Considering the case of autologous cell transplantation, the cost of tissue harvesting and preparation of clinical-grade products, including cell culture, various tests for infections, and labor costs, is expected to be at least several thousand dollars. In the US, the cost of stem cell therapy is USD 4000–8000, and the price of cultured cells is reported to be USD 15,000–30,000 [74]. On the contrary, the average treatment fee for a single dental implant in the US for general dentistry is approximately USD 3000 to USD 4000 [75]. The cost of dental treatments, such as bridges, dentures, and periodontal tissue regeneration, including guided tissue regeneration and bone grafting, is approximately a few thousand dollars, and treatment costs exceeding USD 10,000 are not realistic in dentistry. This compares to diseases that severely affect life and quality of life, such as spinal cord injury, graft-versus-host disease, stroke, and myocardial infarction, in which it may be worth spending over USD 10,000. Currently, stem cell therapy is very expensive compared with regular dental treatments.

## 4. Strategies for Solving Problems

### 4.1. Allogeneic Cell Transplantation

The problems of qualitative and quantitative limitations of stem cells and the cost of treatment considered in autologous cell transplantation may be solved to some extent by allogeneic cell transplantation. The use of cells from young donors allow for the transplantation of a large number of cells with a high therapeutic efficacy. In the case of using stem cells from dental tissues, tissue donations are readily available because most patients with premolar or wisdom teeth extraction are young, and these teeth are often discarded as medical waste. The safety issues associated with allogeneic cell transplantation need to be confirmed in clinical studies in humans. Dhote et al. transplanted allogenic umbilical cord-derived MSCs into human periodontal defects and reported improved periodontal tissue formation and safety after 6 months [59]. Tsumanuma et al. also reported in dogs that transplantation of allogenic MSC from periodontal ligament resulted in the regeneration of cementum and periodontal ligament without any side effects at 8 weeks post transplantation [39]. Further safety verification may solve this safety issue in allogeneic cell transplantation. It is necessary to accumulate sufficient evidence to determine whether the benefits outweigh the drawbacks of autologous cell treatment.

### 4.2. Cell-Free Treatment

The basic idea of tissue regeneration by MSC transplantation is to utilize the differentiation ability of stem cells and to expect that the transplanted stem cells will grow and differentiate locally to regenerate the tissues. However, in many stem cell transplantation studies, the local retention of transplanted cells has been much lower than expected, and it is clear that tissue regeneration and wound healing promotion are mediated by factors secreted from the cells [64,65,66,67]. Secreted factors produced by MSCs are known to have anti-inflammatory, immunoregulatory, angiogenic, and anti-apoptotic functions [76,77]. Based on the potential of secreted factors, tissue regeneration using secreted factors/substances from stem cells is being investigated as a cell-free treatment.

#### 4.2.1. Conditioned Medium

A method has been reported in which secreted factors released from stem cells are collected in the form of culture supernatant and transplanted instead of the cells themselves. Nagata et al. showed that the transplantation of a conditioned medium (CM) obtained from PDLSCs resulted in the regeneration of periodontal tissues. This study demonstrated that PDLSC-CM contained various growth factors and angiogenic factors and that the gene expression of tumor necrosis factor (TNF)-alpha was suppressed at the CM implantation site [78]. Qiu et al. transplanted the CM from gingival MSCs and PDLSCs into periodontal defects in rats, observed that both CMs enhanced periodontal regeneration, and showed that the expression levels of interleukin-1 and TNF-alpha were reduced by CM transplantation using immunostaining [79]. Similarly, periodontal tissue regeneration by the transplantation of CM from bone marrow-MSCs has also been reported [80]. These results indicate that the transplantation of MSC-derived CM induces periodontal regeneration mainly through its anti-inflammatory effect, suggesting the possibility of cell-free treatment using stem cell-derived CM. Animal studies of cell-free treatment for periodontal regeneration are listed in Table 3.

#### 4.2.2. Exosomes/Extracellular Vesicles

Exosomes/extracellular vesicles are small particles (<100 nm) from cells with a lipid bilayer structure. Exosomes contain various cellular components, such as proteins, mRNAs, DNA, and miRNAs. Exosomes released from cells have recently been recognized as new tools for cell–cell communication, as exosomes can be taken up by distant cells and can exert their functions within those cells [81,82]. MSC-derived exosomes contain molecules that modulate wound healing and are believed to be responsible for many of the actions of MSC humoral factors, including angiogenesis, anti-inflammation, anti-apoptosis, and immune regulation [83,84]. Two reports have shown that periodontal tissues can be regenerated by transplantation of exosomes from MSCs. Chew et al. showed that the transplantation of bone marrow MSC-derived exosomes into rat periodontal defects enhanced regeneration of periodontal tissues [85]. Wu et al. reported that the implantation of SHED-derived exosomes induced periodontal regeneration in rats [86].

#### 4.2.3. Features of Cell-Free Treatment

Cell-free treatment, consisting of a CM and exosomes, has features that cover some of the disadvantages of cell transplantation. In CM and exosome transplantation, the possibility of tumor formation and immune rejection is considered to below. In addition, because cells are not transplanted, problems with donor aging and cell sources can be avoided. CMs and exosomes can be stored in freezers, which makes them more practical than cells, as they are inexpensive and easy to manage and use. However, since tissue regeneration with cell-free treatment improves the patient’s wound healing process, the amount of tissue regeneration may be lower than that with cell transplantation. The number of regenerative studies using cell-free treatment is still small compared with that of cell transplantation, and further extensive research is needed.

## 5. Conclusions

Recently, clinical studies on periodontal tissue regeneration by stem cell transplantation in humans have been reported. In three of these studies, there was a trend toward improvement of clinical parameters in the stem cell transplantation group; however, no statistically significant differences were observed [60,62,63]. Autologous cells were used in these studies, and the results of the studies differed from the results of previous animal studies. The quality of transplanted stem cells due to patient aging and chronic inflammation or changes in the transplantation site may have affected the results of human clinical studies. The efficacy of stem cell transplantation for periodontal regeneration has been shown mainly in animal experiments, and we need to find a way to solve the remaining problems to establish the stem cell therapy as a new periodontal regenerative strategy in human patients. Considering the cost and safety issues, much more research needs to be conducted to make stem cell therapy practical for periodontal regeneration. To achieve this, it is necessary to optimize the technique, including the appropriate cell type, number of cells, and graft carrier, and to verify the therapeutic effect and safety through rigorous clinical studies with appropriate controls. Furthermore, the mechanism of periodontal tissue regeneration by stem cell transplantation, including the fate of transplanted cells, needs to be elucidated. Cell-free treatment also requires detailed mechanisms, verification of therapeutic effects, and clinical studies in humans.

Stem cell transplantation studies over the past two decades have provided a great deal of useful information on periodontal regeneration. The information revealed by many stem cell transplantation studies is expected to result in the development of new periodontal tissue regeneration therapies.

## Figures and Tables

**Table 1 ijms-23-01011-t001:** List of preclinical studies for cell-based periodontal regeneration.

Cell Type, Author (Year)	Animal	Scaffold	Observation Period	Results	Regenerated Tissues
BM-MSC					
Kawaguchi et al. (2004) [10]	Dog	Collagen gel	4 weeks	BM-MSC induced more periodontal regeneration than scaffold alone.	CM, PDL, AB
Hasegawa et al. (2006) [11]	Dog	Collagen gel	4 weeks	GFP-labelled transplanted cells were found in regenerated tissues.	CM, PDL, AB
Weng et al. (2006) [12]	Dog	Calcium alginate	12 weeks	BM-MSC increased bone regeneration in periodontal defects.	AB
Li et al. (2009) [13]	Dog	Collagen membrane	8 weeks	Cryopreserved BM-MSC induced periodontal regeneration.	CM, PDL, AB
Wei et al. (2010) [14]	Dog	-	6 weeks	After BrdU-BM-MSC transplantation, some osteoblasts and fibroblasts were BrdU^+^.	CM, PDL, AB
Yang et al. (2010) [15]	Rat	Gelatin beads	3 weeks	Transplanted GFP-BM-MSC were integrated in new CM, PDL, and AB.	CM, PDL, AB
Tsumanuma et al. (2011) [16]	Dog	b-TCP, PGA	8 weeks	CM was thicker in PDLSC defects compared with APC and control.	CM, PDL, AB
Simsek et al. (2012) [17]	Dog	PRP	8 weeks	BM-MSC+PRP showed periodontal regeneration.	CM, PDL, AB
Zhou et al. (2012) [18]	Dog	PLGA	6 weeks	BM-MSC with OPG overexpression enhanced periodontal regeneration.	CM, PDL, AB
Cai et al. (2015) [19]	Rat	PLGA/ɛ-caprolactone	6 weeks	Chondrogenic induction of BM-MSC increased periodontal regeneration.	CM, PDL, AB
Nagahara et al. (2015) [20]	Dog	b-TCP/collagen	8 weeks	b-TCP enhanced AB formation by BM-MSC without affecting CM and PDL regeneration.	CM, PDL, AB
Paknejad et al. (2015) [21]	Dog	Bio-Oss^®^(ABBM)	8 weeks	BM-MSC regenerated more CM and PDL than scaffold.	CM, PDL, AB
Liu et al. (2016) [22]	Dog	collagen-HA	24 weeks	BM-MSC + collagen/HA induce new CM, PDL, and AB formation.	CM, PDL, AB
PDLSC(PDL)					
Nakahara et al. (2004) [23]	Dog	Collagen sponge	4 weeks	More CM was regenerated in PDL-transplanted defects than in empty defects.	CM
Akizuki et al. (2005) [24]	Dog	Hyaluronic acid sheet	8 weeks	Periodontal regeneration found 3/5 defects after PDL cell sheet transplantation.	CM, PDL, AB
Hasegawa et al. (2005) [25]	Rat	-	4 weeks	PDL cell sheet transplantation created new CM-PDL structures.	CM, PDL
Flores et al. (2008) [26]	Rat	-	5 weeks	PDL cells with osteogenic differentiation regenerated more CM.	CM, PDL, AB
Liu et al. (2008) [27]	Minipig	HA/TCP	12 weeks	PDLSC regenerated periodontal tissues lost by ligature-induced chronic inflammation.	CM, PDL, AB
Iwata et al. (2009) [28]	Dog	HA/b-TCP	6 weeks	Triple layered PDL sheet-induced periodontal tissue regeneration.	CM, PDL, AB
Ding et al. (2010) [29]	Minipig	HA/TCP	12 weeks	Allogenic and autologous PDLSC regenerated induced periodontal tissues.	CM, PDL, AB
Park et al. (2011) [30]	Dog	-	8 weeks	PDLSC induced more CM, PDL, and AB than DPSC and periapical follicular stem cells.	CM, PDL, AB
Suaid et al. (2011) [31]	Dog	Collagen composite	12 weeks	PDLSC + GTR membrane promoted periodontal regeneration in class II furcation.	CM, PDL, AB
Tsumanuma et al. (2011) [16]	Dog	b-TCP, PGA	8 weeks	CM was thicker in PDLSC defects compared with APC and control.	CM, PDL, AB
Nunez et al. (2012) [32]	Dog	Collagen sponge	12 weeks	PDL and cementum derived cells regenerated new connective attachment.	CM, PDL
Suaid et al. (2012) [33]	Dog	Collagen composite	12 weeks	PDLSC + GTR membrane promoted periodontal regeneration in class III furcation.	CM, PDL, AB
Mrozik et al. (2013) [34]	Sheep	Gelfoam	4 weeks	Allogenic PDLSC + gelfoam induced CM, PDL, and AB in dehiscence defect model.	CM, PDL, AB
Menicanin et al. (2014) [35]	Sheep	Gelfoam	8 weeks	Autologous PDLSC regenerated periodontal tissues with Shapey’s fiber structure.	CM, PDL, AB
Fu et al. (2014) [36]	Minipig	HA/TCP	12 weeks	Allogenic PDLSC and SHED transplantation resulted in periodontal regeneration.	CM, PDL, AB
Han et al. (2014) [37]	Rat	Gelatin sponge	4 weeks	Allogenic PDLSC transplantation resulted in CM, PDL, and AB regeneration at day 21.	CM, PDL, AB
Iwasaki et al. (2014) [38]	Rat	Amniotic membrane	4 weeks	PDLSC on amniotic membrane induced periodontal regeneration.	CM, PDL, AB
Tsumanuma et al. (2016) [39]	Dog	b-TCP, PGA, collagen	8 weeks	CM regeneration was significant in allogenic PDLSC transplantation without side effects.	CM, PDL, AB
Iwasaki et al. (2019) [40]	Rat	Amniotic membrane	4 weeks	Engraftment of transplanted PDLSC was limited in regenerated periodontal tissues.	CM, PDL, AB
DPSC					
Park et al. (2011) [30]	Dog	-	8 weeks	PDLSC induced more CM, PDL, and AB than DPSC and periapical follicular stem cells	CM, PDL, AB
Khorsand et al. (2013) [41]	Dog	Bio-Oss^®^ (ABBM)	8 weeks	DPSC induced more CM and PDL than the scaffold group. No difference in AB.	CM, PDL
Cao et al. (2015) [42]	Minipig	-	12 weeks	DPSC with HGF overexpression induced more periodontal tissues than DPSC.	CM, PDL, AB
ADSC					
Tobita et al. (2013) [43]	Dog	PRP	8 weeks	ADSC transplantation showed more CM, PDL, and AB formation.	CM, PDL, AB
Ozasa et al. (2014) [44]	Dog	Bolheal^®^ (fibrin gel)	6 weeks	ADSC induced new CM, PDL, and AB formation.	CM, PDL, AB
Venkataiah et al. (2019) [45]	Minipig	Bolheal^®^ (fibrin gel)	4 weeks	Allogenic ADSC regenerated periodontal tissues, comparable with autologous ADSC.	CM, PDL, AB
APC					
Jiang et al. (2010) [46]	Dog	b-TCP	12 weeks	APC+b-TCP improved periodontal regeneration in class III function.	CM, PDL, AB
Tsumanuma et al. (2011) [16]	Dog	b-TCP, PGA	8 weeks	CM was thicker in PDLSC defects compared with APC and control.	CM, PDL, AB
GMSC					
Fawzy El-Sayed et al. (2012) [47]	Minipig	Collagen	12 weeks	GMSC-transplanted sites demonstrated better CAL, PD, GR, and HAL than the control.	CM, PDL, AB
Yu et al. (2013) [48]	Dog	-	8 weeks	GFP-labelled GMSC integrated in regenerated tissues.	CM, PDL, AB
SHED					
Fu et al. (2014) [36]	Minipig	HA/TCP	12 weeks	Allogenic PDLSC and SHED transplantation resulted in periodontal regeneration.	CM, PDL, AB

AB: alveolar bone, ABBM: anorganic bovine bone matrix, ADSC: adipose tissue-derived stem cells, APC: alveolar bone periosteal cells, BM: bone marrow, CAL: clinical attachment level, CM: cementum, DPSC: dental pulp stem cells, GFP: green fluorescent protein, GMSC: mesenchymal stem cells from gingival connective tissue, GR: gingival recession, HA: hydroxyapatite, HAL: histological attachment level, HGF: hepatocyte growth factor, MSC: bone marrow derived-mesenchymal stem cells, OPG: osteoprotegerin, PD: pocket depth, PDL: periodontal ligament, PDLSC: periodontal ligament stem cells, PGA: polyglycolic acid, PLGA: poly-lactide-co-glycolide acid, SHED: stem cells from human exfoliated deciduous teeth, TCP: tricalcium phosphate.

**Table 2 ijms-23-01011-t002:** Results of human controlled clinical trials for cell-based periodontal regenerative therapy.

Author (Year)	Cell Type	Sample Size	Experimental Groups	Observation Period	Statistical Significance
Dhote et al. (2015) [59]	Allogenic Cord MSC	14 patients 24 defects	Control: open flap debridement Test: beta-TCP + rhPDGF-BB + cells	6 months	◯ (CAL gain, PPD reduction, radiographic bone fill)
Chen et al. (2016) [60]	Autologous PDLSC	Total 41 Control 21 Test 20	Control: GTR +BioOss Test: GTR + PDLSC + BioOss	3, 6, 12 months	× (CAL, PPD, GR)
Ferrarotti et al. (2018) [61]	Autologous DPSC	Total 29 Control 14 Test 15	Control: Collagen sponge Test: Collagen sponge + cells	6, 12 months	◯ (CAL gain, PPD reduction, radiographic bone defect fill)
Sánchez et al. (2020) [62]	Autologous PDL-MSC	Total 19 Control 9 Test 10	Control: Xenogeneic bone substitute Test: Xenogeneic bone substitute + cells	6, 9, 12 months	× (Test group showed greater CAL gain and PPD reduction than control with no statistical significance)
Apatzidou et al. (2021) [63]	Autologous BM-MSC	Total 27 Each group 9	Group A: collagen + fibrin/platelet lysate + cells Group B: collagen + fibrin/platelet lysate Group C: Flap surgery	12 months	× (All group showed similar results)

MSC: mesenchymal stem cells, PDLSC: periodontal ligament stem cells, PDL: periodontal ligament, BM: bone marrow, TCP: tricalcium phosphate, PDGF: platelet-derived growth factor, GTR: guided tissue regeneration, CAL: clinical attachment level, PPD: probing pocket depth, GR: gingival recession.

**Table 3 ijms-23-01011-t003:** List of animal studies of cell-free treatment for periodontal regeneration.

Author (Year)	Cell Free Treatment	Cell	Animal Model	Results
Nagata et al. (2017) [77]	CM	PDLSC, dermal fibroblasts	rat	PDLSC-CM enhanced periodontal regeneration and inhibited TNF-alpha expression. PDLSC-CM contained various growth factors, angiogenic factors, and cytokines. CM from dermal fibroblasts did not induce regeneration.
Qiu et al. (2020) [78]	CM	PDLSC, GMSC, gingival fibroblasts	rat	CMs from PDLSC and GMSC induced periodontal regeneration. TNF-alpha and IL-1beta levels were lower in these CM-transplanted sites.
Kawai et al. (2015) [79]	CM	BM-MSC	rat	BM-MSC-CM contained IGF-1, VEGF, and TGF-beta. CM transplantation enhanced periodontal regeneration.
Chew et al. (2019) [84]	Exosome	BM-MSC	rat	Exosome enhanced proliferation and migration of PDL cells. Exosome transplantation promoted periodontal tissue healing.
Wu et al. (2019) [85]	Exosome	SHED	rat	SHED-exosome increased angiogenic gene expression in endothelial cells and osteogenesis-related genes in BM-MSC through AMPK. SHED-exosome transplantation increased bone formation in rat periodontal defect model.

AMPK: adenosine monophosphate-activated protein kinase, CM: conditioned medium, GMSC: gingival mesenchymal stem cells, IGF: insulin-like growth factor, IL: interleukin, PDLSC: periodontal ligament stem cell, SHED: stem cells from human exfoliated deciduous teeth, TGF: transforming growth factor, TNF: tumor necrosis factor, VEGF: vascular endothelial growth factor.

## Data Availability

Not applicable.

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
