# Peer review of "Stem Cell Transplantation and Cell-Free Treatment for Periodontal Regeneration"

_ijms, 2022, doi:10.3390/ijms23031011_

Round 1

Reviewer 1 Report

General comment
1. This manuscript will be helpful to researchers by presenting previous research results, limitations, and future research on stem cell transplantation and cell-free treatment for periodontal regeneration.
2.
I recommend presenting the previous research results on cell-free treatment as a 'table' so that many researchers can easily compare various previous studies.

Author Response

We are grateful to the reviewer for the constructive comment. Following the suggestion from the reviewer, we summarized the previous studies about cell-free treatment as Table 3 in the revised manuscript. Corrections are shown in red in revised manuscript.

Thank you for your constructive comment.

Reviewer 2 Report

This review highlights the major problems and solutions regarding stem cell transplantation therapy for periodontal regeneration. The manuscript is clear, precise, easy to understand, and offering potentially important information. However, the following concerns should be addressed before the manuscript can be considered for publication:

- Line 23-26 –Please, rewrite the paragraph.

- The authors should highlight the composition of the periodontal tissue.

- The authors should include in Table 1 the results observed for each study.

- For each cell transplantation study presented in Table 1, the authors should describe which tissue was regenerated (alveolar bone, cementum, periodontal ligament).

- Line 138-140 – Please add some references:

            - Carvalho MS, Alves L, Bogalho I, Cabral JMS, da Silva CL. Impact of Donor Age on the Osteogenic Supportive Capacity of Mesenchymal Stromal Cell-Derived Extracellular Matrix. Front Cell Dev Biol. 2021;9:747521. doi:10.3389/fcell.2021.747521.

- Line 148 – Please, correct “10^6 to 10^7” to “106 to107”.

Author Response

We would like to express our appreciation to the reviewer for valuable comments. Our responses to each comment are mentioned below. All corrections are shown in red in revised manuscript.

  1. Rewrite of the paragraph (line 23-26)

Thank you for your suggestion. We paragraphed line 23 to 26 in revised manuscript.

  1. Composition of periodontal tissue

Thank you very much for this constructive comment. We agree that the composition of periodontal tissue is very important in understanding periodontal tissue regeneration. We have added a sentence on this point in the revised manuscript (line 34-35).

  1. Results and regenerated tissues in each study in Table 1

Thank you for your suggestion. Following the constructive comment from the reviewer, we added summary of the results of each study and the regenerated tissues to Table 1.

  1. Reference regarding stem cell aging

Thank you very much for your suggestion. We added the suggested reference as #71 in the revised manuscript.

  1. The description of powers

We corrected the description of powers, following the suggestion from the reviewer (line 156).